# A Mixed Methods Study to Implement the Synergy Tool and Evaluate Its Impact on Long-Term Care Residents

**DOI:** 10.3390/healthcare11152187

**Published:** 2023-08-02

**Authors:** Farinaz Havaei, Francis Kobekyaa, Andy Ma, Maura MacPhee, Wei Zhang, Megan Kaulius, Bahar Ahmadi, Sheila Boamah, Adam Easterbrook, Amy Salmon

**Affiliations:** 1School of Nursing, The University of British Columbia, Vancouver, BC V6T 2B5, Canada; 2Faculty of Pharmaceutical Sciences, University of British Columbia, Vancouver, BC V6T 1Z3, Canada; 3Centre for Health Evaluation & Outcome Sciences, Vancouver, BC V6Z 1Y6, Canada; 4School of Nursing, McMaster University, Hamilton, ON L8S 4K1, Canada; 5Faculty of Medicine, The University of British Columbia, Vancouver, BC V6T 1Z3, Canada

**Keywords:** long-term care, workplace management, Synergy Model

## Abstract

Background: There are ongoing workforce challenges with the delivery of long-term care (LTC), such as staffing decisions based on arbitrary standards. The Synergy tool, a resident-centered approach to staffing, provides objective, real-time acuity and dependency scores (Synergy scores) for residents. The purpose of this study was to implement and evaluate the impact of the Synergy tool on LTC delivery. Methods: A longitudinal mixed methods study took place within two publicly funded LTC homes in British Columbia, Canada. Quantitative data included weekly Synergy scores for residents (24 weeks), monthly aggregated resident falls data (18 months) and a six-month economic evaluation. Qualitative data were gathered from family caregivers and thematically analyzed. Results: Quantitative findings from Synergy scores revealed considerable variability for resident acuity/dependency needs within and across units; and falls decreased during implementation. The six-month economic evaluation demonstrated some cost savings by comparing Synergy tool training and implementation costs with savings from resident fall rate reductions. Qualitative analyses yielded three positive impact themes (*improved care delivery*, *better communication*, and *improved resident-family-staff relationships*), and two negative structural themes (*language barrier* and *staff shortages*). Conclusions: The Synergy tool provides useful data for enhancing a ‘fit’ between resident needs and available staff.

## 1. Introduction

Long-term care (LTC) homes in Canada serve high-risk and vulnerable older populations, with the majority living with complex and multiple chronic comorbidities. The assumption that LTC home residents are stable with low needs is no longer applicable [1,2]. Cognitive impairments and dementia rates are rising amidst overcrowding and understaffing challenges at LTC homes, including a lack of appropriate staffing levels and training to meet the increasingly acute and complex needs of residents [3,4,5]. These issues were exacerbated by the COVID-19 global pandemic, which had a devastating and deadly impact on residents, family caregivers (FCGs), and staff [1,5,6,7,8]. In addition, cultural and ethnic diversity is on the rise among residents in Canadian LTC homes, especially in urban areas. Ethnocultural diversity can have an impact on the quality and safety of care delivery, such as communications with staff [9].

Despite increases in LTC residents’ physical, cognitive, and emotional-social needs over recent decades, LTC staffing decisions are often based on arbitrary standards and economic reasons rather than residents’ needs [8,10]. The same staffing models and provider hours per resident per day (HRPD) have remained static for decades, implying a presumption that resident needs do not vary across individuals, units, care homes, and over time. In fact, for the last 20 years, HRPD standards in Canada have been based on US data from 15,000 care homes in 10 States [11]. Depending on the resident case mix, which is a US measure of acuity, the 2001 recommendations were for a range of 2.4–2.8 HRPD for care aides (CAs), 1.15–1.30 HPRD for licensed or registered practical nurses (LPNs or RPNs depending on jurisdiction), and 0.55–0.75 HPRD for registered nurses (RNs), totaling 4.1–4.85 h of direct care provider daily per resident. Since then, US recommendations have largely prevailed without further examination of residents’ needs in Canadian LTC homes. In many instances, LTC homes from some provinces in Canada do not meet the 2001 recommended standards. For example, the average HPRD in Ontario was 3.73 in 2018, and 3.28 HPRD in British Columbia in 2020 [12,13].

Non-profit advisory groups in Canada regularly report on HPRD staffing averages and resident outcomes using quality indicators, such as falls with and without injuries (Office of the Senior’s Advocate British Columbia, 2020; Ontario Long-Term Care Staffing Study Advisory Group, 2020). Lower HPRD standards in Canadian LTC homes no doubt compromise basic resident-centered care needs and outcomes [10,14]. Research evidence is beginning to link a richer skill mix of regulated nurse (RNs and LPNs/RPNs) HPRD with improved resident outcomes, including decreased falls, urinary tract infections, pressure ulcer occurrence and weight loss; and improved pain management [15,16,17,18].

In Canada, there is limited literature on current resident care needs and effective staffing levels and skill mix to ensure safe, quality care delivery. From a resident-centered care perspective, it is necessary to know residents’ acuity and dependency needs to staff according to these needs. Some recent evidence suggests that one resident assessment tool, the Synergy tool, can provide real-time information about individual residents’ priority care needs, to determine an appropriate staffing complement. The conceptual Synergy Model and its accompanying patient needs assessment tool (Synergy tool) were developed in the 1990s in the US to assist nurses with objectively assessing and quantifying their patients’ acuity and dependency needs. Acuity refers to patient needs that are overseen by regulated nurses, and dependency needs include activities of daily living (e.g., bathing, feeding, ambulating) that are delivered by unregulated CAs or non-nurse professionals (e.g., nutritionists, physio-occupational therapists). Psychometric evaluation of the Synergy tool for acute care and specialty settings was carried out in the US where the tool was widely adopted over the last 20 years [19]. Subsequently, the conceptual model and tool spread internationally, including in Canadian jurisdictions [20,21,22].

Research with the Synergy tool has demonstrated its capacity to facilitate safe staffing and workload management decisions in real-time by using health professionals’ assessments of patients’ acuity and dependency needs. The ultimate purpose of the Synergy tool is to enhance the quality and safety of care delivery by creating a better ‘fit’ between staffing assignments and patient needs [20,21,23,24]. Research evidence suggests that Synergy tool use leads to the optimization of existing health human resources [20,21,23], positive care delivery experiences and outcomes for patients and providers, and reduced costs in healthcare delivery [22,23]. The majority of research with the Synergy tool has been in acute care, emergency services and specialty and ambulatory care programs [20,21,22,23,25,26,27,28,29]. To our knowledge, research with the Synergy tool has not been conducted in the LTC sector. In addition, there has never been any research containing an economic evaluation of Synergy tool implementation, despite the importance of cost analysis for guiding operational and policy-level decisions [30,31,32]. The purpose of this study was to implement and evaluate the impact of the Synergy tool on residents’ care delivery in two ethnically diverse LTC homes in large urban areas within British Columbia, Canada, including evaluation from an economic perspective.

## 2. Materials and Methods

### 2.1. The Synergy Tool Intervention 

The generic Synergy tool consists of eight patient characteristics [20,24]. There are five acuity characteristics—complexity, vulnerability, resilience, stability, and predictability—and three dependency characteristics that include resource availability and capacity to make decisions and perform activities of daily living [23,24]. After a professional does a primary patient assessment, they score the patient on a Synergy tool five-point scale to differentiate between high (1–2) moderate (3) and low (4–5) needs for each of the tool’s eight characteristics [20,21]. To assist with scoring, the Synergy tool scale is accompanied by assessment indicators developed by expert healthcare providers that distinguish between high-moderate-low needs for each acuity and dependency characteristic. Details about the Synergy tool with its scale and assessment indicators can be found in a published generic toolkit [33]. Individual ‘Synergy scores’ highlight key patient needs priorities, and they can be aggregated or averaged to examine changes in patient care needs over different time intervals and/or across different patient groups [20]. Individual and aggregated Synergy scores can also be correlated with administrative and survey data.

### 2.2. Intervention Implementation

When introducing the Synergy tool as an intervention in any care setting, facilitated two-day workshops are held with expert healthcare team members who best know their patient population. For this study, we organized a two-day virtual workshop via Zoom videoconferencing with 20 LTC senior direct care staff and their clinical leadership as well as family and/or resident caregiver representatives from two partner LTC homes. The workshop agenda and activities were guided by the published generic toolkit [33]. Key workshop activities included: (1) adaptation of the generic Synergy tool scale and assessment indicators for residents in LTC homes (see Appendix A); and (2) staff training and reliability determination using resident case scenarios from the two homes. As an example of determining resident indicators utilizing the adapted Synergy tool, individuals that require total care and constant supervision would be classified as “high need” in the Participation in Care indicator, whereas individuals that are one-person assistance and need support for some, but not all, activities of daily living would be considered “moderate”, and individuals that are able to be independent in most activities of daily living would be considered “low”. A plan for implementation and data collection for the study was also determined collaboratively with the workshop participants. Two units from each LTC home with different resident needs were purposefully selected for the Synergy tool implementation, which occurred between June and November 2022. The scoring teams consisted of three expert nurses from one home and seven from the second home. On a weekly basis, scorers used the Synergy tool to score each resident in their home. To do so, scorers used documents, such as resident charts and care plans; they had conversations with direct care staff and family caregivers; and in many instances, they did collaborative assessments with direct care staff. There were, therefore, resident-centered and educative components to the implementation. In addition, scorers relayed scores to charge nurses for revising staff assignments as needed, and if status changes were noted, scorers collaborated with direct care staff to notify appropriate providers, such as medical officers for the LTC homes and/or family caregivers as needed. There were some instances when additional resources or float nurses were assigned to specific residents due to the identification of concerning status changes.

### 2.3. Study Design

An explanatory sequential mixed methods research design [34] was employed in two ethnically diverse LTC homes in large urban areas within British Columbia, Canada. The two care homes have a bed size ranging from approximately 150 to 200 publicly funded beds with 100% and nearly 50% of their residents being of Chinese and Jewish ethnocultural background respectively, with numerous dialects (e.g., Mandarin, Cantonese). The project goal was to gain a comprehensive understanding of the impact of the Synergy tool implementation on residents’ care delivery in these LTC homes and to address the following research questions: (1) How does the use of the Synergy tool capture LTC resident acuity and dependency needs in real time and over time? (2) What evidence from administrative data supports the effectiveness of the Synergy tool with respect to resident outcomes? (3) What is the economic impact of the Synergy tool for LTC homes? (4) What evidence from family caregivers suggests that resident care delivery improved during Synergy tool implementation? This study was approved by the Research Ethics Boards of the University of British Columbia, the two partner LTC homes and/or their respective health authority (H22-00772).

### 2.4. Quantitative Methods

To answer our first research question, we collected and analyzed residents’ Synergy scores; and to answer our second research question, we used resident falls data from LTC homes’ administrative databases. The third research question was examined by comparing the cost of a Synergy tool implementation with the cost savings associated with resident fall reduction after Synergy tool implementation.

#### 2.4.1. Sample and Data Collection

During the six-month implementation period, scoring of all residents on the four units by the trained scorers took place between June 2022 to November 2022, and Synergy scores for each resident were recorded on a digital extraction form developed by the researchers. Assessments were carried out for all residents on a weekly basis on different days of the week and at different times in a 24-h period. Overall, 24 weeks of Synergy scores were collected for 85 unique residents across four units. To enable individual data linkage over time, each set of Synergy scores contained a resident’s unique anonymized identifier, and their level of need (i.e., high, moderate, low) for each of the eight Synergy characteristics.

Aggregated resident falls data for the four units was extracted and de-identified from the administrative databases of the two LTC homes. The extracted resident falls indicator was aggregated at a unit level and was a rate indicator constructed from the number of resident falls divided by the number of residents in a given unit each month. Data were collected on a monthly basis for 18 months to obtain twelve months of data before Synergy tool implementation (June 2021 to May 2022) and six months of data during the implementation (June 2022 to November 2022).

#### 2.4.2. Data Analysis

Data analysis consisted of descriptive statistics and data visualization: Visualization methods were unique components of this study. To visualize the variability of Synergy scores across both residents and time, grid heat maps were created, with rows representing individual residents, columns representing time by week, and rows grouped by unit [35]. The acuity or dependency needs for each resident were calculated by averaging their respective characteristics scores, resulting in a comprehensive overview of resident needs over time. Another visualization approach of unit-level broken-line graphs overlaid upon resident-level smoothed lines (created through cubic spline interpolation) [36] was used, to compare unit-level means with resident-level variability.

For the administrative data, line graphs were used to descriptively illustrate the change in resident falls rates across 18 months, with a two-part linear trendline depicting the pre-Synergy tool implementation and during implementation trends in falls rates overall. Trendlines were created by retrieving predicted intercepts and slopes from an interrupted time series model. The interrupted time series model was fitted using mixed-effect modeling, in which initial intercepts were permitted to vary across units.

For the economic evaluation, the cost of the Synergy tool implementation was estimated based on the training workshop and labor input incurred during this study and then allocated among residents. The cost savings from reducing the fall rate (number of resident falls per resident) were valued by the healthcare cost associated with fall-related incidents as previously estimated in extant literature.

Inferential statistics were not appropriate for the study design, as the administrative falls data had limited sample sizes. The administrative falls data was aggregated by unit, resulting in a sample size equivalent to the number of implementation units (n = 4). As such, while statistical models were used to extract overall intercepts and slopes for visualization and rough estimation purposes, the intent of their usage was not to test for the statistical significance of estimates. *p*-values are presented for transparency only.

### 2.5. Qualitative Methods

To answer our fourth research question, we conducted a qualitative descriptive study to obtain the ‘voice’ of those with lived experience [37]. Another purpose for collecting qualitative data was to enable better triangulation of findings from the quantitative data [38,39].

#### 2.5.1. Sample and Data Collection 

We recruited a convenience sample of English-speaking family members (n = 6) from the two LTC homes to gain their perspectives on the quality of their loved ones’ care delivery. Given the high dementia prevalence among residents in both LTC homes, with support from leadership, we recruited family members in caregiving roles for residents who visited the home at least three times per week.

Two focus groups were conducted (one per home). One focus group included two family members and a follow-up interview was conducted with another family member who was unavailable for their scheduled focus group. The other home’s focus group consisted of three family members. A semi-structured guide consisted of the following questions: (1) Have you noticed any changes in your (e.g., mother’s/father’s) care delivery over the last six months? If so, can you describe the changes? (2) Have you noticed any changes in the staff’s ability to meet your (e.g., mother’s/father’s) care needs over the last six months? If so, can you describe the changes? An experienced qualitative researcher facilitated the focus group and interview sessions, ensuring that all participants had opportunities to share their perspectives in an open and transparent fashion.

Data collection was conducted virtually and facilitated by FH, FK, and BA. Separate 60–90-min focus groups and an interview were conducted with family caregivers over a period of four weeks towards the end of the six-month Synergy tool implementation period between September and October 2022. The focus groups and interviews were audio-recorded, transcribed verbatim using Zoom transcription software (version 5.11.11), anonymized, accuracy checked, and uploaded to QSR NVivo R.1.6 version for data analysis.

#### 2.5.2. Data Analysis

Qualitative content analysis [40] using both within-case (within LTC site) and cross-case (across LTC sites) analyses were conducted. The one interview was included with the focus group transcript of the LTC home for the interviewee’s resident family member. Coding was completed by three investigators (FK, MM, FH) who did consensus-checking at regular intervals. This process included coding the raw data line-by-line and developing interpretations about recurring, converging and contradictory codes or ideas within and across the data. This process allowed data-driven themes to be constructed and interpreted, providing a rich description of the participants’ points of view [40].

## 3. Results

### 3.1. Quantitative Study 

#### 3.1.1. Synergy Scores

Weekly Synergy scoring was conducted for a total of 85 unique residents. The average count of Synergy scores per resident was 18.6 (SD = 7.1), due to factors such as resident unavailability during weekly scoring and bed changes. The mean resident age was 87.4 (SD = 8.0) and the resident sex was 62.4% female.

We used two unique methods to visually display individual resident needs by unit over time (Research Question 1). The grid heat maps in Figure 1 and Figure 2 showed clear differences in patterns of resident needs between units. For resident Acuity needs, units 1A and 2A showed low overall needs with stable individual trajectories across time; unit 2B showed a mix of low, moderate, and high-need residents with stable individual trajectories; while unit 1B showed higher variability in needs, both between residents and across time. For resident Dependency needs, unit 2B showed stable individual trajectories for their residents, who were mostly low needs; unit 1A showed stable trajectories for low and moderate needs residents; unit 2A showed slightly variable trajectories for residents spanning low, moderate, and high needs; and unit 1B showed highly variable trajectories for residents who were moderate and high needs. These heat map findings indicate that units 1A and 2B had more stable, lower-needs residents; unit 2A had residents with low acuity needs and a diverse mix of dependency needs, with slight individual variability across time; and unit 1B had residents with rapidly varying needs.

Figure 3 is another visualization strategy of the same acuity and dependency data displayed in Figure 1 and Figure 2 (Research Question 1). The light blue lines in Figure 3 depict individual resident acuity and dependency variability over time, and the solid, colored lines represent the average of resident needs in a single unit over time.

Figure 4 further highlights the unit-level differences in resident needs score variability (Research Question 1). For each resident, an absolute difference was created per week, by taking the absolute value of the difference between one week’s score and the previously available score. Resident absolute differences were averaged by unit per week, highlighting the changes in resident needs over time. Overall, Unit 1B displays higher week-by-week variability in resident needs scores. A sharp jump in resident needs score changes occurred for Unit 1A in week 26. Jumps in score changes over time can pinpoint when leaders need to more closely examine reasons for data dips and peaks and the required supports and resources to address changing needs.

#### 3.1.2. Administrative Resident Falls Data

The aggregated resident falls rate indicator was calculated per implementation unit, with a numerator of the number of resident falls in that unit that month and a denominator of the number of residents in the unit (Research Question 2). Figure 5 shows that in the year prior to the Synergy tool implementation, the overall linear trend as averaged across all four units (intercept = 0.186, *p* = 0.04; slope = +0.006, *p* = 0.23) displayed an increase in falls rates. The interrupted time series model estimated the intercept reduction at the start of the implementation to be −0.083 (*p* = 0.22), and the slope reduction to be −0.005 (*p* = 0.74). At the onset of the implementation phase, the counterfactual trend was therefore extrapolated to begin at 0.261 in June 2022, and increase at a rate of 0.006 per month. In comparison, the overall estimated trend was estimated to begin at 0.174 in June 2022, and increased from there at a slower overall rate (slope = +0.001) than the pre-implementation counterfactual trend. However, due to the limited sample size of four units, neither the level change nor the slope change were statistically significant.

#### 3.1.3. Economic Evaluation

For pragmatic purposes, we based our economic evaluation on training and implementation costs for an ‘average Canadian’ LTC home. The average Canadian LTC home size is 153 residents and 15 residents/beds per unit [41]. With respect to staff training and scoring, we calculated costs based on the standard training approach described in the generic toolkit. Table 1 provides a detailed breakdown of training and implementation/scoring costs (Research Question 3). This cost analysis does not account for potential staff turnover and ongoing scorer training.

The cost of a potential six-month implementation in a 150-bed care home was estimated at $67.01 per resident. The cost savings associated with the reduction in resident falls was roughly estimated using the extrapolated pre-implementation falls rate trend and the implementation falls rate trend from this study. At the 12-month point in the administrative data period, the extrapolated pre-implementation (counterfactual) trend was calculated as 0.261 falls/resident, with an increase of 0.006 falls/resident per month following. The implementation trend was calculated as 0.174 falls/resident, with an increase of 0.001 falls/resident per month following. Averaged across a six-month implementation period, the counterfactual rate was 0.277 falls/resident, and the implementation rate was 0.176 falls/resident; indicating a falls rate reduction of 0.101 falls/resident per month, or a reduction of 0.606 falls/resident across a six-month implementation period. A review of previous studies on the LTC-related treatment cost per resident fall showed that the potential median cost per resident fall was $12,504.4/fall in 2022 Canadian dollars after adjusting for inflation using the Consumer Price Index for health and personal care [45,46]. Accordingly, the potential cost savings associated with falls rate reduction was estimated at $7573.47 per resident for a six-month implementation.

### 3.2. Qualitative Study

#### 3.2.1. Demographic Characteristics of Participants

We recruited family caregivers (n = 6) from two LTC homes in BC. Participants were between the ages of 55 and 70 years old. Of these, 4 were female and 2 were male; 4 were children, one was a spouse, and one was a sibling of a resident.

The interpreted themes were based on majority comments and agreement among and across the two focus groups and the one interviewee. Some selective quotes from the focus groups are presented with themes below. There were three positive impact themes (*improved care delivery*, *better communication*, *and improved resident-family-staff relationships*). There were two negative structural themes (*staff shortages* and *language barrier*) that pertained to existing structural barriers versus participant observations of negative impacts from Synergy tool Implementation (Research Question 4). They are included as the contextual background of LTC barriers that hamper safe, quality care delivery.

#### 3.2.2. Positive Impacts Themes

##### Improved Care Delivery

Residents’ family caregivers reported that they observed the scorers using the Synergy Tool to perform individual resident assessments. They noted how discussions between scorers and staff often included more than technical tasks, such as ways to provide personalized care (often including them in the discussions). Given the ethnic diversity of residents, one family member stated:

*“I do notice that, well, in the last few months [name of care home withheld] is a culturally oriented care home, which we so appreciate because of our heritage”* [LTC2].

This family member also observed special efforts to tailor care to each resident’s specific needs.

*“...so they do allow for that kind of individualized kind of attention and care [to residents] and that I really appreciate.”* [LTC2]

As stated by another caregiver:

*“For example, every time when I visited my mom, even though, they’re still updating me what’s going on… they are always very accommodating, even though my mom’s health is deteriorating, it’s natural. And when I sent my mom back at [night]time there, you know, all the staff came down to take my mom in. So just little things showing that the staff really, really care.”* [LTC2]

With respect to safety, one family member observed more attention to safety features, including discussions between scorers and staff about their family member’s high risk for falls.

*“But they’re doing one fine job… keeping everyone safe.”* [LTC2]

##### Better Communication

According to the participants, communications with them improved with respect to the quality of communications, or the messaging about their loved ones.

*“There is much more communication about that now [from staff], that relieves me a bit …”*. [LTC1]

*“…and I appreciate the fact that they really keep me informed and yesterday, they called to tell me he [the resident] has decided to go for a little stroll”*. [LTC2]

Participants provided instances of inclusive messaging to ensure their presence in care coordination and care planning discussions.

*“...I really appreciate the service and care that [name of care home withheld] is giving to my mom. I think, previously I already had three care meetings, and each of the individuals responsible for that department will report what’s happening to my mom. I’m very touched and moved by all the care…”.* [LTC2]

##### Improved Resident-Family-Staff Relationships

Focus group participants reported examples of increased staff efforts to involve them in residents’ social activities—to enhance the social experiences for their loved ones.

*“I often see, you know, some of the residents sitting in the TV room, and there’s a nurse sitting right beside them, or a nurse assistant…… there are some sort of engagements, or they’re like, hey, come, sit here and let’s chat!”*. [LTC1]

*“It’s impossible to show you right here, but the connection between the family, the volunteer and the residents is really quite strong. And this is really, really positive, I think.”* [LTC2]

#### 3.2.3. Negative Structural Themes

##### Staff Shortage

All the focus group participants reported observations where residents did not get their needs met in a timely fashion.

*“I’m finding perhaps there’s less, not as often attention to the care, because sometimes they’re short-staffed where there’s a new staff, and they’re not aware of the routine, and I understand, because of Covid, staffing shortages, there’s not as much care as there needed to be.”* [LTC2]

*“But I do realize that this is not the ideal world. I do realize there are staffing issues that I’m concerned about.”* [LTC2].

One family member voiced:

*“I don’t blame them. They’re busy. They can’t stand around …. But I understand the limitations of staffing, the limitations of space and doesn’t know how it works that way.”* [LTC1]

##### Language Barrier

Participants described how language barriers are a key challenge for non-English speaking residents. Given the importance of communication, family members observed that healthcare providers could not engage in effective conversation with these residents without family members available to translate.

*“My mother can understand a little bit of English, but I do notice it’s a little bit hard for her because it creates a bit anxiety and nervousness because she does not understand what the staff is trying to tell her…I think it creates problems for the staff themselves having to do interpretations. So, it is a little difficult”*. [LTC2]

*“With the shortages or lack of speaking Cantonese in the native language, it’s difficult to interact with them”*. [LTC2]

*“His [respondent’s husband] first language, I believe, is Hebrew. He’s just found out he doesn’t speak English [any longer]”*. [LTC1]

## 4. Discussion

Our study demonstrated how the Synergy tool can be used in real-time and over time to capture and operationalize the acuity and dependency needs of individual residents, and to provide an overall average of acuity and dependency needs for specific units or resident populations over time. These findings reflect what healthcare providers and literature have been saying about our resident populations. Advanced aging with accompanying physical, cognitive and social-emotional conditions means that every resident will have different healthcare needs to consider when delivering resident-centered care. In addition, seniors, especially the baby boomer generation, have strong preferences for alternative living accommodations, and they expect more control over the quality and types of services they receive [47]. Resident-centered care models that acknowledge and respond to individual variability in resident needs are also associated with decreased behavioral symptoms and antipsychotic medication use in residents with dementia [48]. As stated by family caregivers in the focus groups and interviews, attention to residents’ specific needs was of great value to them. We were unable to include residents in our focus groups due to the high dementia rates in our two LTC homes, but one US study found that cognitively intact residents appreciated many aspects of resident-centered care delivery, particularly waking and bedtime choices, having consistent staff, and having a voice to discuss concerns and to make changes via resident councils. Residents also noted how their ‘homes’ had a long way to go to de-institutionalize their living environments [49].

The Synergy tool is a resident-centered assessment tool that can be easily used by healthcare professionals to collaboratively plan and personalize care with residents and their family caregivers. As evidenced by our Synergy score data, there can be considerable individual variability in LTC residents’ acuity and dependency needs, and on some units, those needs can change daily. In our data, one unit (Unit 1B) had acuity needs similar to medical-surgical acute care units in another research we have conducted [20]. Our findings have dispelled the assumptions that LTC homes have stable residents with non-changing basic care needs. The Synergy scores can inform creative ways of utilizing the funded HPRD based on resident needs, such as by assigning the more experienced, regulated nurse to high-acuity residents. Likewise, Synergy scores can be used by researchers to evaluate the validity of the funded HPRD for quality and safe resident care delivery.

In Canadian LTC homes, residents are assessed using the Resident Assessment Instrument-Minimum Data Set 2.0 (RAI-MDS 2.0) questionnaire, which is completed on admission and quarterly. The questionnaire contains over 300 items, making it difficult to use for real-time assessments. Several problems have been identified by the developers of the RAI-MDS, such as its medical/acuity orientation and its inability to assess characteristics that are quickly changeable over time [50]. Absent from the questionnaire are questions about the quality of life, resident autonomy, satisfaction and level of dependency [51]. Some Canadian provinces, such as BC, have created resident/family experience surveys, but these surveys are administered and reported through the Seniors’ Advocate annually [52]. In a qualitative, descriptive study of RAI-MDS, researchers’ interviews with RAI-MDS coordinators who collect data in their LTC homes described the data as “decontextualized” and a “click box of predetermined items” (Armstrong et al., 2017, p. 359). The collected data, by trained coordinators, is sent to the National Canadian Institute of Health Information (CIHI), cleaned, synthesized and sent to provincial seniors advocate offices. Due to the nature of data collection and reporting, it is highly unlikely that data can be used to efficaciously inform the decisions of providers, residents, and their family caregivers [51]. In contrast to current resident assessment tools in place, such as the RAI-MDS, the Synergy tool characteristics and assessment indicators can be quickly adapted and validated by expert healthcare professionals in the LTC home and implemented in real time and as frequently as deemed appropriate by the expert providers.

In our study, we used a quality indicator from LTC homes’ administrative data, the falls rate, to determine if the use of the Synergy tool had any impact on resident outcomes. Falls rates are a quality indicator reported to CIHI as part of the RAI-MDS, and for licensing and accreditation purposes, LTC homes need to track falls [50]. Falls are common in residents for a number of reasons, including physical and cognitive impairments that affect perception, balance and coordination; and medications that increase the risk of falls. Other factors include urinary incontinence (e.g., trying to reach the bathroom quickly), and disinhibited, risk-taking behaviors associated with certain mental health conditions [53]. In a 2015 Dutch study that recorded reasons for falls in one nursing home, there was considerable variability related to factors influencing falls in residents. Over a 19-month observational period, 85% of the residents had falls, and about 30% of these falls had serious consequences, such as broken hips. The researchers concluded that preventively, it is important to identify those individuals at the highest risk for falls and to provide more effective, individualized fall prevention [53]. Our preliminary declining falls rates during the study’s implementation period suggest that the Synergy tool has the potential to detect and intervene with high fall-risk residents.

We were unable to locate any recent LTC cost analyses in published literature. Acute care cost analyses typically focus on quality indicator outcomes and compare how different interventions result in cost savings from decreased adverse outcomes. One cost analysis study conducted in a US Acute Care for Elderly (ACE) unit, for example, demonstrated how specialized care for bedbound, frail elderly in an ACE unit resulted in significant hospital cost savings from decreased length of hospital stay and unplanned hospital readmissions, and increased functional capacity at the time of discharge [54]. Cost analyses, therefore, should be an integral component of intervention evaluation, especially given finite healthcare resources. Based on published acute care economic evaluations, we used documented fall rates, a quality indicator, to determine the cost-effectiveness of Synergy tool implementation. Our economic evaluation yielded promising results, although it was a six-month period of time and it is unknown if the Synergy tool impact would extend to other resident outcomes, such as pressure ulcers [54].

Based on our qualitative data, we know that family caregivers observed and sometimes participated in conversations about individual residents’ care needs. Synergy scoring helped to raise awareness of residents with specific needs or safety risks, such as the potential for falls, and to explore and confirm subtle changes in resident status with family caregivers. Our qualitative findings are congruent with research conducted in emergency services with the Synergy tool, where staff stated that the use of the Synergy tool made them more aware of the holistic needs of each patient, especially overlooked needs, such as psychosocial needs [25,26]. These studies also found that nurses were more apt to proactively contact other healthcare team members to address other care needs concerns, such as social care needs for vulnerable individuals with housing and food insecurity issues.

The qualitative findings of our study reinforce the importance of resident-centered approaches that promote information-sharing and care decision-making between staff, residents, and families. Other themes pertained to ongoing LTC quality of care structural barriers: staff shortages and language barriers. The Synergy tool may play a role in addressing these structural barriers. With respect to staff shortages, during the implementation period, the scorers at both LTC homes, who were regulated professionals, communicated frequently with direct care staff (e.g., CAs) and family members to more accurately assess the needs of individual residents. In many ways, the pattern established between scorers, staff, and residents/families was an example of team-based care. Team-based care is considered one of the most efficient and effective modes of care delivery, and research has shown how this care delivery approach can improve patient outcomes and even decrease provider burnout [55]. Team-based care, however, rarely happens in LTC homes due to the predominant number of unregulated staff in direct care roles. Given the ongoing nursing shortage, it is unlikely that there will be a significant uptake of regulated nurses within LTC homes. Team-based care, therefore, will need to center on CAs working in collaboration with available regulated nurses (RNs, LPN/RPNs), family caregivers and residents. Our qualitative findings provide feasibility for an LTC approach to team-based care that optimizes the knowledge of residents that CAs possess.

Over a decade ago, health researchers in Canada predicted the need to recognize the growing importance of CAs as a valuable health human resource [56]. In addition to carrying out activities of daily living (e.g., toileting, bathing, feeding, ambulating), CAs are assuming nurse-related tasks delegated to them [57]. In the Synergy tool training workshop, we did within the LTC homes, we included CAs in the workshop to invite their perspectives on important assessment indicators to include with the Synergy scoring process. Their knowledge of the residents contributed to the overall implementation process, and although scoring was completed by expert regulated professionals, the CAs were integral to the development of resident assessment indicators that accompany the Synergy tool. Including CAs in Synergy tool training and scoring discussions, therefore, may be one significant way of building a greater sense of team and enriching the care planning process for each resident.

Given the rising ethnocultural diversity in residents of LTC homes, the Synergy tool can be used to identify residents and family members with special needs pertaining to their unique ethnic and cultural identities. Communication difficulties for non-English speaking residents are common care delivery barriers in LTC [9]. Some published literature has addressed LTC language barriers from the staff perspective where English is a second language for many CAs [58]. As globalization continues, all healthcare services will need to reckon with language barriers due to English-second language providers and residents. Our qualitative findings, however, provided examples of how consistent staff and knowledge of residents and their caregivers resulted in more personalized, and culturally sensitive care. CAs worked with family caregivers to know residents’ cultural needs and to provide services adapted to those needs. In other healthcare sites, such as emergency services and acute care, professional interpreter services are commonly utilized, and interpreters can also act as cultural brokers [59]. In under-resourced LTC homes that lack professional interpreter services, the relationships forged with families by CAs often act as proxies for overcoming language barriers. The Synergy tool can act as a means to capture CA-family knowledge of ways to better meet the diverse ethnocultural needs of individual residents.

### Limitations

To our knowledge, this is the first implementation and evaluation study of the Synergy tool in the LTC context. Despite its novelty, the study has key weaknesses. First, due to structural constraints such as LTC staffing shortages and resource inadequacies, Synergy scoring of residents happened only when scorers had available time. To avoid the likelihood of bias, ideally, scorers should have scored assigned residents at a set time each shift after resident assessment. Second, we would like to acknowledge that during the Synergy tool implementation, other interventions were taking place in the participating units, perhaps contributing to the declining falls rates. In other words, the declining fall rates might not be completely attributable to the Synergy implementation, and neither were the associated cost savings. However, considering its low cost, the Synergy implementation could be still cost-effective or cost-saving. Third, the intervention took place in two care homes within a specific geographic location; as such, findings should be cautiously generalized to LTC contexts other than the partner care homes.

## 5. Conclusions

To conclude, qualitative and quantitative evidence from this study suggests that the Synergy tool holds promise as a means of improving care delivery, communication, and resident-family-staff relationships by providing simple, real-time opportunities to assess and collaboratively determine residents’ unique acuity and dependency needs. Data can be used in real-time for identifying individual resident needs and any status changes. Over time (e.g., daily, weekly, quarterly), leadership can use Synergy score trends to better understand staffing needs, on average, for each unit of care. Visual trends also reveal peaks and dips in data that require further exploration. The power of the cost-effective Synergy tool is its capacity to inform resident-centered care approaches within LTC homes.

## Figures and Tables

**Figure 1 healthcare-11-02187-f001:**
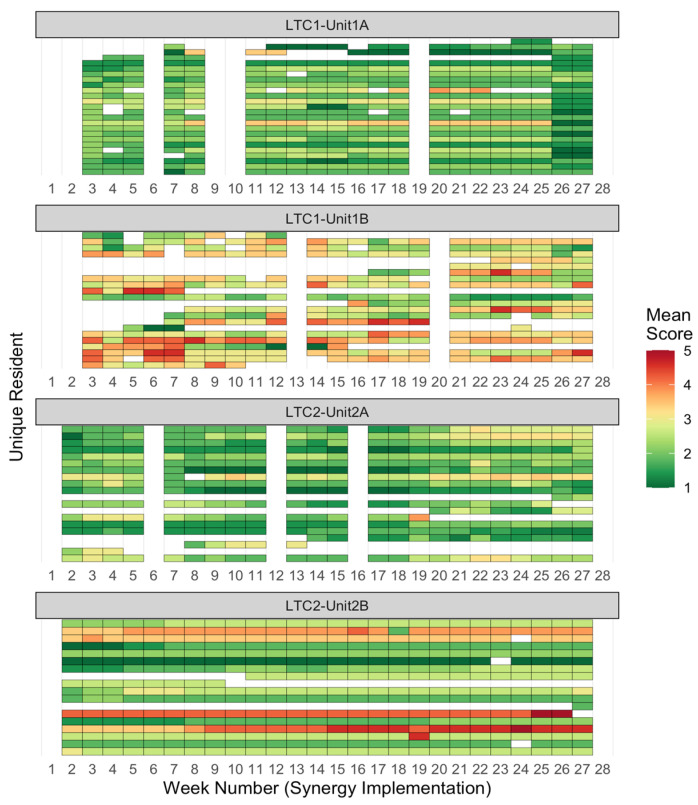
Mean Score Heatmap for Acuity Needs per Resident, by Week. Note. Each colored cell represents one resident’s Acuity Needs score for that week. An Acuity Needs score is the average of a resident’s Stability, Complexity, Predictability, Resiliency, and Vulnerability needs. Each row of cells represents one resident’s needs scores over time; each column, one week of scores. Higher mean scores are colored red and signify higher needs.

**Figure 2 healthcare-11-02187-f002:**
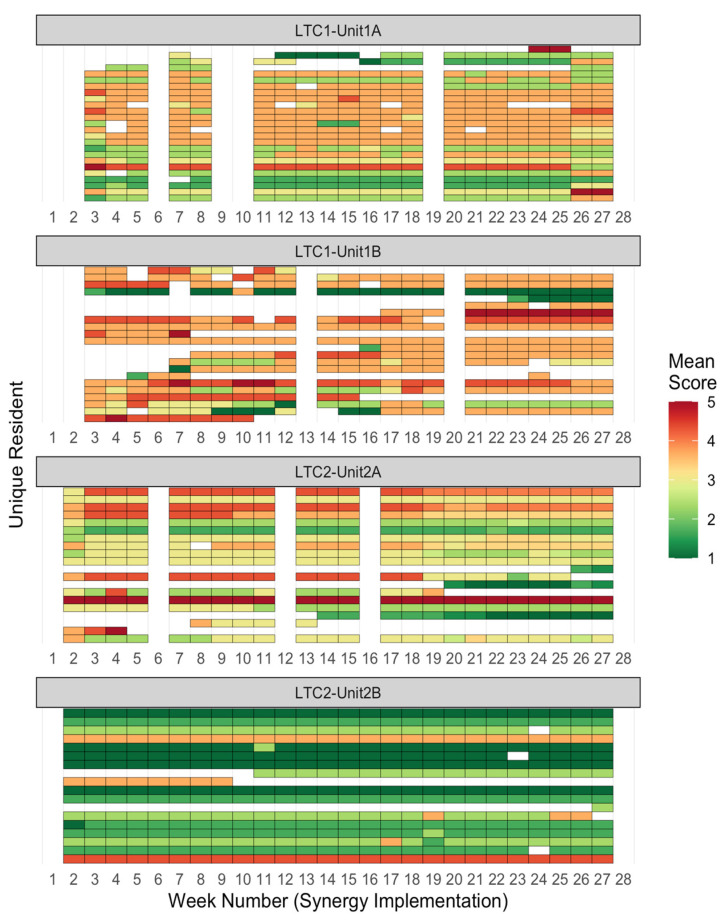
Mean Score Heatmap for Dependency Needs per Resident, by Week. Note. Each colored cell represents one resident’s Dependency Needs score for that week. A Dependency Needs score is the average of a resident’s Participation in Decision, Participation in Care, and Resource Availability needs. Each row of cells represents one resident’s needs scores over time; each column, one week of scores.

**Figure 3 healthcare-11-02187-f003:**
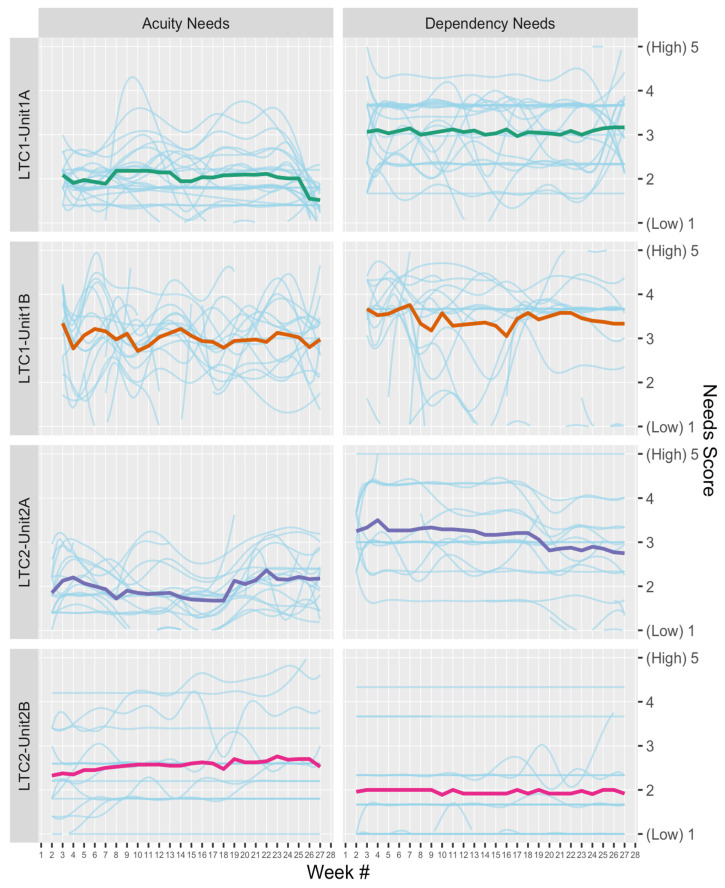
Acuity and Dependency Scores Over Time. Note. Each plot contains a unit-level broken-line that depicts unit averages for resident acuity/dependency scores, as well as resident-level cubic spline interpolated lines that visualize the range and volatility of resident needs in a unit.

**Figure 4 healthcare-11-02187-f004:**
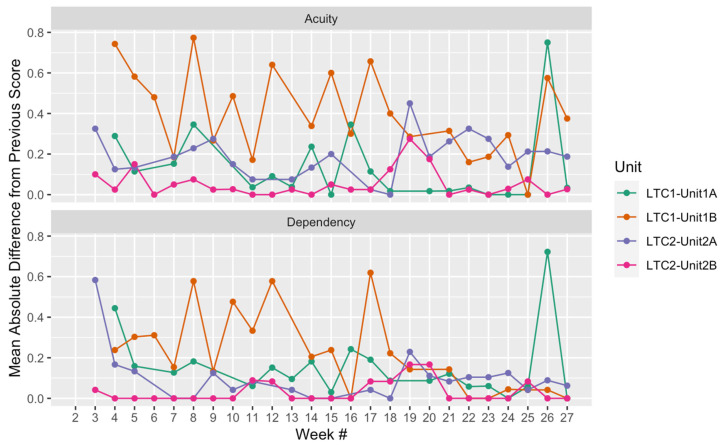
Unit-level Averages for Absolute Differences from Previous Scores.

**Figure 5 healthcare-11-02187-f005:**
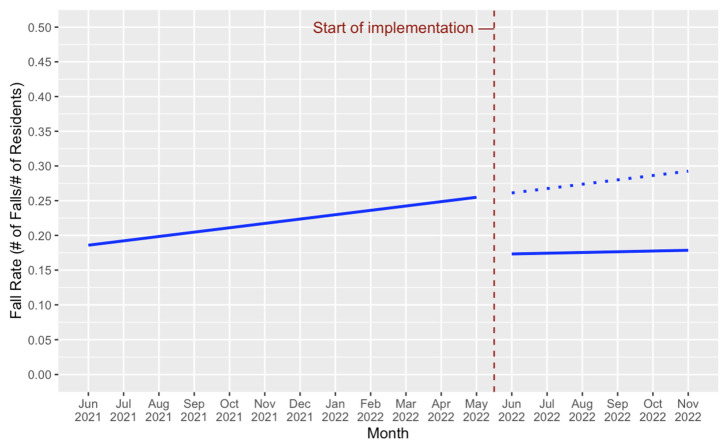
Overall Trend in ‘Fall Rate’ Resident Indicator. Note. The pre- and during intervention overall trend in fall rate is displayed by the solid blue line. Extrapolation of the pre-intervention trend is shown by the dotted blue line.

**Table 1 healthcare-11-02187-t001:** Cost estimate breakdown, highlighting the six-month cost of a potential Synergy implementation in a 150-bed LTC home.

Cost Component	Description	Estimate Breakdown	Cost Estimate
Training workshop	Release time for staff to participate in training, using the average of median hourly rates for care aides [42], LPNs/RPNs ^1^ [43], and RNs [44] ($33/h), factoring in an approx. benefit of 25%, and training 4 scorers per 15-bed unit.	$33/h	$6600
		×1.25 (benefits)	
		×4 h	
		×4 scorers/unit	
		×10 units	
	Honoraria ($100 per person) for two family/resident representatives, to provide insight for minor refinement of indicators during training	$100/person	$200
		×2 reps.	
	Facilitation costs, for two expert facilitators ($60/h)	$60/h	$480
		×4 h	
		×2 facilitators	
Post-workshop support	Weekly office hours held by one facilitator in the month following the workshop, to address scoring questions	$60/h	$90
		×30 min/wk.	
		×3 weeks	
Scoring time	Costs for staff to score residents on a weekly basis during implementation; Synergy Tool scoring requires approximately one minute of time per resident	$33/h	$2681.25
		×1.25 (benefits)	
		×1 min/week	
		×150 residents	
		×26 weeks	
Total cost of six-month implementation (150-bed LTC home)			$10,051.265
Cost per resident for a six-month implementation			$67.01

Note: ^1^ Licensed Practical Nurses (LPNs) in British Columbia are equivalent to Registered Practical Nurses (RPNs) in Ontario, Canada.

## Data Availability

Data are available upon reasonable request and approval from the Research Ethics Boards of the University of British Columbia and partner care homes.

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
