# Peer review of "A Mixed Methods Study to Implement the Synergy Tool and Evaluate Its Impact on Long-Term Care Residents"

_healthcare, 2023, doi:10.3390/healthcare11152187_

Round 1

Reviewer 1 Report

Dear authors, thank you for let me reviewing your work titled “A mixed methods study to implement the Synergy Tool and evaluate its impact on long-term care residents”.

I see it as a mixed study (quantitative and qualitative) to implement and assess the “Synergy Tool” in long term institutionalized residents.

Due to the facts of the increasing aging  and the Covid impact, in the long term care institutions, any effort to improve it deserves to be showed to the scientific community.

Let me please, to do some considerations and questions.

Considerations:

In introduction, there are repeated paragraphs from 99 to 163 lines.

The last results paragraph (lines 472 to 474) looks like a “reviewer coment”, that could really be applied to the “discussion” section.

I see the figures like a tools to have information with a visualization, but It seems difficult to me understand the concept “Error! Reference source not found”. 

 Questions:

I wonder why the qualitative data were gathered only from family caregivers and not to workers too. I suggest to explain in discussion or considering it as a study limitation. 

Why authors only measure “falls”, despite other indicators like urinary tract infections, pressure ulcer ocurrence, weight or improved pain management, when are mentioned in introduction (lines 66-67).

In line 541-43 author state that “…Synergy tool has the potential to detect… fall-risk residents. But, how it is dectected? Could authors explain it?

Reviewer 2 Report

Dear Authors,

the text has numerous editorial problems, repetitions of entire paragraphs, it is written chaotically in fragments. We lack broader information about the components of the Synergy system, those that are then presented on maps and charts. What is included in Acuity Needs per Resident? How is Stability, Complexity, Predictability, Resiliency, and Vulnerability needs calculated?

The qualitative part of the work is described freely. No synthesis. The cultural element has been reduced to two linguistic remarks. All remarks are marked in the attached text.

Overall, though, an interesting idea, probably well executed. But it is necessary to organize the text.

Reviewer 3 Report

Thank you for the opportunity to review this manuscript, extensive and very well constructed throughout. I have two comments. Can you reassure me that the use of focus groups was related only to a family situation and did not include non family members and if this is the case how effective was a focus group in the provision of encouraging family members to speak openly? Was there power imbalances? Also there are a small number of lines eg 281/310/334/343 that have Error Reference not found can this be updated please.

Reviewer 4 Report

In this manuscript, the authors implemented and evaluated the impact of the Synergy tool on the provision of care for residents of two different ethnic groups in long-term care homes in the metropolitan area of British Columbia, Canada, including evaluating the Synergy tool from an economic perspective. The author demonstrated through qualitative and quantitative analysis that Synergy tools can provide a simple and real-time opportunity to evaluate and collaborate to determine residents' unique sensitivity and dependency needs, thereby improving care delivery, communication, and relationships with residents' families and staff. I would suggest accepting it after the following major concerns are addressed.

1. Will the increasing cognitive impairment and dementia rates in the population with long-term care adaptation mentioned in the research background affect the research results due to the different conditions of patients? The inclusion and exclusion criteria were not mentioned after the article.

2. Due to the limited sample size, which may affect the statistical results, it is recommended to increase the sample size.

3. When describing statistical methods, it is necessary to indicate why the fall rate is used as a quality indicator in LTC household management data.

4. Because it is not a strict randomized Scientific control, this study cannot explain that the results must be affected by Synergy tools.

5. The author's discussion section did not mention future research directions.

Minor editing of English language required
